# Monocytes from neonates and adults have a similar capacity to adapt their cytokine production after previous exposure to BCG and β-glucan

Rhoda Namakula[1], L. Charlotte J. de Bree[2], Tor Henrik A. Tvedt[3], Mihai G. Netea[2], Stephen Cose[4,5], Kurt Hanevik[6,7]*

**1** Centre for Intervention Science in Maternal and Child Health, Centre for International Health, University of Bergen, Bergen, Norway, **2** Department of Internal Medicine and Radboud Center for Infectious Diseases, Radboud University Medical Center, Nijmegen, Netherlands, **3** Department of Medicine, Section for Hematology, Haukeland University Hospital, Bergen, Norway, **4** Department of Clinical Research, London School of Hygiene and Tropical Medicine, London, United Kingdom, **5** MRC/UVRI and LSHTM Uganda Research Unit, Entebbe, Uganda, **6** Department of Clinical Science, Faculty of Medicine, University of Bergen, Bergen, Norway, **7** Department of Medicine, Norwegian National Advisory Unit on Tropical Infectious Diseases, Haukeland University Hospital, Bergen, Norway

* kurt.hanevik@med.uib.no

**Data Availability Statement:** All data from the present study are available from the NSD database (dx.doi.org/10.18712/NSD-NSD2722-V1).

## Abstract

The Bacillus Calmette-Guérin (BCG) vaccine is administered at birth in tuberculosis (TB) endemic countries. BCG vaccination is also associated with protective non-specific effects against non-tuberculous infections. This seems at least in part mediated through induction of innate immune memory in myeloid cells, a process termed *trained immunity*. β-glucan, a component of the fungal cell wall from *Candida albicans*, induces a trained immunity phenotype in human monocytes with hyper-responsiveness against unrelated pathogens. We aimed to study the capacity of BCG and β-glucan to induce a similar phenotype by examining cytokine production in cord blood monocytes following re-stimulation. We used a well-known model of *in vitro* induction of trained immunity. Adherent mononuclear cells from neonates and adults, which consist mainly of monocytes, were stimulated in vitro with BCG or β-glucan for one day, after which the stimulus was washed away. Cells were rested for 5 days, then restimulated with LPS. Cytokine levels were measured using ELISA. Neonate and adult monocytes responded similarly in terms of cytokine production. BCG significantly increased IL-6 responses to LPS in both neonate and adult monocytes, while β-glucan induced increases of IL-6, IL-10 and TNF production capacity. The BCG and β-glucan induced increase in cytokine production, reminiscent of trained immunity, showed similar levels in neonatal and adult monocytes. BCG mediated changes in cytokine production shows the feasibility of this *in vitro* assay for further studies regarding non-specific effects of vaccines.

**Funding:** This work was supported by research group funds from the University of Bergen, 3593288 (Mrs Rhoda Namakula) and Haukeland University Hospital. M.G.N was supported by a Spinoza Grant of the Netherlands Organization for Scientific Research and an ERC Advanced grant (#833247).

**Competing interests:** The authors have declared that no competing interests exist.

## Introduction

In accordance with WHO recommendations, Bacillus Calmette-Guérin (BCG) vaccination is normally given at birth in tuberculosis (TB) endemic countries to protect against severe forms of TB [1]. Furthermore, BCG vaccination has also been suggested to promote additional non-specific beneficial effects on overall child survival [2, 3]. However, the mechanisms by which BCG may cause this non-specific protective effect is not clear, but recent studies point towards trained immunity as a potential mechanism [4]. Trained immunity is a general enhanced long-term responsiveness of innate immune cells to microbial stimuli mediated by monocytes and natural killer (NK) cells [4]. β-glucan, a component of the fungal cell wall from *Candida albicans*, has shown to induce training of human monocytes with increased responsiveness not only against fungi, but also against bacteria, viruses and even parasites [4]. The mechanism behind this has been by functional, transcriptional and epigenetic reprogramming through a dectin-1/ Raf1-dependent pathway [5, 6].

Monocytes are innate immune cells and an important source of cytokine production during microbial infections. They express a range of pattern recognition receptors, including high expression of Toll-like receptors (TLRs) [7]. Newborns are more vulnerable than adults to infections partly due to limited antigenic experience and to the premature status of adaptive immunity in newborns [8], placing a greater burden on innate immunity for host defense to microbial challenge in young infants [9]. Increased monocyte-derived cytokine responses lasting several months after BCG vaccination have been demonstrated upon *in vitro* stimulation of peripheral blood mononuclear cells (PBMCs) or whole blood with unrelated pathogens or TLR agonists in both infants and adults [10–12].

Upon exposure to certain microbial ligands, human monocytes have been shown to undergo trained immunity, both *in vivo* and *in vitro*, via a mechanism of epigenetic reprogramming, resulting in increased responsiveness to secondary stimuli by microbial pathogens and increased cytokine production [10, 13].

BCG and β-glucan have been shown to induce trained immunity in adult monocytes [14] in a well-known model of *in vitro* induction of trained immunity. However, little is known about the capacity of neonatal monocytes to maintain immunological information from a previous antigenic experience. In this study, making use of this model [14], we aimed to investigate whether human cord blood adherent monocytes (CBAM) possess, similar to adult peripheral blood adherent monocytes, the capacity to adapt their cytokine production after previous exposure to BCG and β-glucan.

## Materials and methods

### Study participants

In total, 45 participants were recruited. Umbilical cord blood (CB) from 15 newborn full term babies were collected at the maternity ward of Haukeland Hospital, Bergen, Norway. We obtained informed consent from pregnant mothers. Eligibility requirements included pregnant mothers who were to deliver by cesarean section. Mothers were approached and the study purpose explained to them in a language they could understand. Those who gave informed consent on behalf of their newborn babies were recruited in the study. Mothers with known blood transmitted infections were not included in the study. For every neonate, peripheral blood samples from two healthy adults in a volunteer pool of 30 adults were collected. Exclusion criteria were any blood disorders, immunodeficiencies or any medication that would affect the immune system. BCG vaccination status for 27 adult volunteers was known.

Out of these, 22 had been vaccinated, all of them more than 10 years ago, (until 2005 Norwegians were vaccinated at around age 10–14 years) while 5 adults were not vaccinated.

## Ethical considerations

Ethical approval was obtained from the Ethics committee at Haukeland Hospital, Bergen, Norway. All the participants consented to take part in the study.

## Experimental methods

Cord and adult samples were collected in Acid Citrate Dextrose (ACD) tubes, and obtained within a mean 4 (range 3–5) hour timeframe. They were processed simultaneously, under the same conditions.

Placental cord blood mononuclear cells (CBMCs) and adult peripheral blood mononuclear cells (PBMCs) were isolated by Lymphoprep (GE Healthcare, UK) using density centrifugation from blood diluted 1:2 in sterile phosphate-buffered saline (PBS). Cells were washed three times in cold PBS and the pellet was re-suspended in 1ml of Dutch modified Roswell Park Memorial Institute Medium (RPMI-1640) culture medium (Invitrogen, CA, USA) supplemented with 50 μg/mL gentamicin, 20 mM GlutaMAX, and 10 mM pyruvate.

## *In vitro* assay of adherent monocytes

A total of $5 \times 10^5$ mononuclear cells in a 100 μl volume were added to polystyrene flat-bottom 96-well plate (Corning, NY, USA) for 1 hr at 37°C for monocytes to adhere and cells were washed three times with warm PBS to remove non-adherent cells [14]. The adhered monocytes were then cultured at 37°C in RPMI supplemented with 10% pooled sterile human serum. During the first 24 hours cells were primed by being cultured with either live-attenuated BCG Bulgaria (Intervax, Sofia, Bulgaria) (5ug/ml) or the *Candida albicans* (*C. albicans*) cell wall component β-glucan (1ug/ml), a potent inducer of trained immunity which was kindly provided by Professor David Williams (East Tennessee State University, USA) [14], or cultured with RPMI only as negative control. After 24 hours the plates were washed once with 200 μL warm PBS. The culture medium was changed on day 4. On day 6 the cells were re-stimulated with *Escherichia coli* LPS (Serotype O55:B5, Sigma-Aldrich, St. Louis, MO, USA) (10 ng/ml) or left unstimulated. On day 7, supernatants were harvested and stored at −80°C.

## Cytokine measurements

IL-6, IL-10 and TNF were measured in stored supernatants using commercial Duoset ELISA kits (R&D Systems MN, USA) according to the manufacturer's instructions.

## Statistical analyses

Cytokine responses were analyzed by calculating the fold change (measured stimulated cytokine values were divided by the values for the RPMI-only wells. Data were not normally distributed. We therefore, used the Wilcoxon matched-pairs sign-rank-test for comparisons of cytokine responses in trained monocytes to values in the RPMI-only well cultures. Mann-Whitney non-parametric test (Wilcoxon-rank-sum-test) was used to compare cytokine levels between adults and neonates.

The statistical analyses were performed in STATA version 15 (StataCorp. 2009; Stata Statistical Software, College Station TX) and graphs were drawn using GraphPad Prism 7 (Graph Pad Software, Inc., San Diego, CA, USA). A two-tailed p value was considered significant if $< 0.05$.

## Results

Adult volunteers in the study had a mean age of 43 years (range 20–67), with 66% being female. Neonate samples were completely anonymized, thus gender was not available for this group.

### Cytokine profiles to LPS following BCG and β-glucan priming in neonates

We examined cytokine production from CBAM stimulated with BCG or β-glucan, then re-stimulated with LPS on day 6 and compared their cytokine production capacity with unprimed, LPS re-stimulated conditions.

There was a significant increase in IL-6 production following BCG priming in neonates, but TNF and IL-10 production were not different. (Fig 1A). β-glucan priming resulted in a significant increase of all cytokine levels IL-6, IL-10 and TNF tested (Fig 1A).

The middle horizontal lines represent the median; boxes cover 50% of the values between the 25th and the 75th percentiles, with the lower 25% and upper 75% percentiles. Abbreviations: BCG, Bacillus Calmette-Guérin; IL, interleukin; TNF, tumor necrosis factor; LPS, Lipopolysaccharide; RPMI, Roswell Park Memorial Institute medium.

### Cytokine profiles to LPS following BCG and β-glucan priming in adults

Similar to the data for infant cytokine responses, adult peripheral blood adherent monocytes primed with BCG induced increased levels of IL-6 and β-glucan initial stimulation led to significantly increased concentrations of IL-6, IL-10 and TNF (Fig 1B).

### The magnitude of cytokine profiles between neonates and adults

Cytokine responses were generally of similar magnitude in adults and neonates. Cytokine production between neonates and adults following either BCG or β-glucan training was compared for each of the three cytokines. We did not observe significant differences in production of the three cytokines tested following initial stimulation with either BCG or β-glucan.

## Discussion

Monocytes play an important role in BCG-induced trained immunity in adults [10]. Induction of non-specific innate immunological memory is a mechanism proposed to explain non-specific beneficial effects in neonates after BCG vaccination [11].

In the present study we show a change in cytokine production in both neonatal and adult monocytes, with upregulation of IL-6 production in response to LPS after initial BCG and β-glucan exposure.

This is the first study to investigate the capacity of CBAM to respond with altered cytokine production and whether it is to a similar degree as adult monocytes. Our data shows that some CBAM cytokine responses, when re-stimulated with LPS following initial priming with either BCG or β-glucan stimuli, are increased compared to LPS-re-stimulated non-primed CBAM. We also demonstrate that β-glucan can induce changes in cytokine production in CBAM in a way similar to adult adherent monocytes through increased production of the three cytokines tested (IL-6, IL-10, and TNF) after re-stimulation with LPS.

Bekkering *et al*, showed that β-glucan priming increased not only pro-inflammatory cytokine production (IL-6 and TNF) but also anti-inflammatory cytokine (IL-10 and IL-1Ra) production following resting of cells for 6 days. These findings are similar to the current study that shows a significant increase of all cytokines (IL-6, IL-10, and TNF) following β-glucan training.

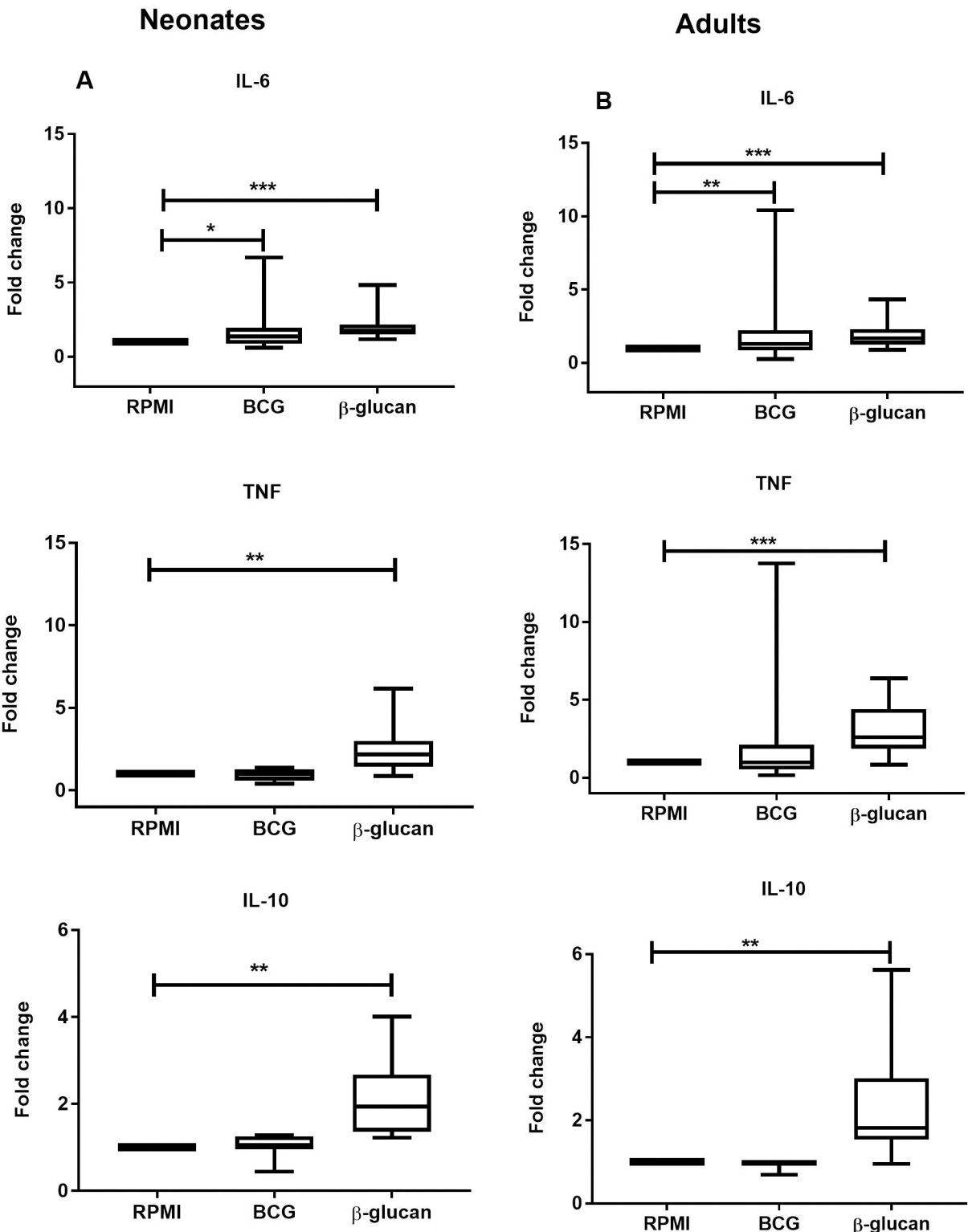

**Fig 1.** Cytokine profiles to LPS following BCG and β-glucan priming as compared to medium control conditions (A) In neonates, IL-6 (n = 15) cytokine production significantly increased in BCG-primed monocytes. (B) In adults, IL-6 (n = 30) production after LPS re-stimulation increased significantly after BCG priming. β-glucan priming of monocytes led to a significant increase of all cytokine levels (IL-6, IL-10 and TNF) in both neonates and adults. $^*P < .05$; $^{**}P < .01$; $^{***}P < .001$.

In a previous study of peripheral blood adherent monocytes from adults, using a similar protocol as conducted in our current study, increased production of TNF following initial BCG training and re-stimulation with LPS was reported [10]. In the current study, the increase in TNF in adults did not reach significance and further comparison is difficult because IL-6 and IL-10 were not measured in the previous study. In the previous studies by Bekkering *et al* and Kleinnijenhuis *et al* BCG Denmark was used, while for the present study BCG Bulgaria strain was used instead, due to production shortages at SSI-Denmark, and lack of availability of BCG-Denmark strain. Several studies have found BCG Denmark to be more immunoreactive than BCG Bulgaria [15, 16]. However, whether this affects the ability to induce alterations in cytokine response is not known. Further studies utilizing the adherent monocyte model could be a feasible *in vitro* model to compare the ability of various BCG strains to induce trained responses.

Other studies have demonstrated increased IL-6, IL-1β, TNF, IL-10 and IFN-γ cytokine responses after BCG vaccination upon later *ex vivo* non-mycobacterial stimulation in both infants and adults [10–12]. In infants, it has been demonstrated that BCG vaccination led to significantly increased levels of IL-6 and IL-10 after *in vitro* whole blood stimulation with non-related microbial stimuli four months later, but not of TNF [11]. These findings suggest that trained immune responses to BCG may last for several months.

A randomized trial of low birth weight infants receiving BCG at birth measured cytokine responses to innate agonists like LPS in whole-blood assays after 4 weeks. BCG vaccination was associated with increased cytokine levels, particularly of IL-1β, IL-6, TNF, and IFN-γ, but no significant effect was detected at the level of IL-10 production [12]. These results concur with our current *in vitro* results after BCG priming, except that we did not see a significant rise in TNF levels.

A number of previous studies have revealed the epigenetic and metabolic mechanisms underlying the increased responsiveness in trained immunity using the present model [10, 14, 17]. Such further characterization of the broad trained immunity phenotype of cord blood adherent monoctes was beyond the scope of the present study, and was not assessed. However, further studies may look into potential differences in these mechanisms between neonates and adults.

Although there are differences between umbilical cord blood and venous blood from adults, adherent cells were isolated in the same way. It should be noted that non-immune cells, such as mesenchymal and hematopoietic stem cells, are present in umbilical cord blood and have adhesive properties. Such cells have shown the potential to be trained [18]. However, they are very few in number and therefore unlikely to cause bias in the observed cytokine profile. A study by Normann *et al*, showed a profile of CBAM with a purification of non-active but stimulation-competent monocytes with high yields ($2.3–9\times10^7$ cells) and purity of 70–90% (average 83%) by adherence or 90–98% by flow cytometric analysis [19].

We acknowledge that cord blood monocytes may not be equivalent to neonatal peripheral blood monocytes. However, we did not identify any studies that have examined this question and therefore this should be investigated in future studies.

The results of this study indicate that this *in vitro* model of trained immunity is applicable also in neonates and therefore may be helpful in evaluating present and future vaccine candidates with regard to their heterologous innate immune effects. Including measurement of epigenetic and metabolic alterations in such experiments are important to characterize the nature of the effects and mechanisms behind them.

## Conclusion

Our findings show that BCG vaccine is capable of inducing increased cytokine production, reminiscent of trained immunity, in cord blood adherent monocytes. BCG vaccination at birth may therefore modulate the development of the neonatal immune system and may play a role in the non-specific effects of BCG observed in epidemiological studies.

## Acknowledgments

We appreciate the adult volunteers and mothers who consented for their babies to be involved in the study. Thanks to the research colleagues at Haukeland University hospital who assisted with cord blood collection and to Victoria Nankabirwa for useful comments to the manuscript.

## Author Contributions

**Conceptualization:** Mihai G. Netea, Stephen Cose, Kurt Hanevik.

**Data curation:** Rhoda Namakula.

**Formal analysis:** Rhoda Namakula, L. Charlotte J. de Bree, Kurt Hanevik.

**Funding acquisition:** Kurt Hanevik.

**Investigation:** Rhoda Namakula, Kurt Hanevik.

**Methodology:** Rhoda Namakula, L. Charlotte J. de Bree, Tor Henrik A. Tvedt, Mihai G. Netea, Stephen Cose, Kurt Hanevik.

**Project administration:** Rhoda Namakula, Kurt Hanevik.

**Resources:** L. Charlotte J. de Bree, Tor Henrik A. Tvedt, Mihai G. Netea, Kurt Hanevik.

**Software:** Kurt Hanevik.

**Supervision:** Kurt Hanevik.

**Validation:** Rhoda Namakula, L. Charlotte J. de Bree, Mihai G. Netea, Kurt Hanevik.

**Visualization:** Rhoda Namakula, L. Charlotte J. de Bree, Mihai G. Netea, Stephen Cose, Kurt Hanevik.

**Writing – original draft:** Rhoda Namakula.

**Writing – review & editing:** L. Charlotte J. de Bree, Tor Henrik A. Tvedt, Mihai G. Netea, Stephen Cose, Kurt Hanevik.

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
