## [Decision Letter · Decision Letter 0]

21 Aug 2019

PONE-D-19-19049

Monocytes from neonates and adults have a similar capacity to elicit trained immunity responses to BCG

PLOS ONE

Dear Dr Hanevik,

Thank you for submitting your manuscript to PLOS ONE. After careful consideration, we feel that it has merit but does not fully meet PLOS ONE’s publication criteria as it currently stands. Therefore, we invite you to submit a revised version of the manuscript that addresses the points raised during the review process.

We would appreciate receiving your revised manuscript by Oct 05 2019 11:59PM. To enhance the reproducibility of your results, we recommend that if applicable you deposit your laboratory protocols in protocols.io, where a protocol can be assigned its own identifier (DOI) such that it can be cited independently in the future. For instructions see: http://journals.plos.org/plosone/s/submission-guidelines#loc-laboratory-protocols

We look forward to receiving your revised manuscript.

Kind regards,

Angelo A. Izzo

Academic Editor

PLOS ONE

Journal Requirements:

2. Please provide additional details regarding participant consent.

In the ethics statement in the Methods and online submission information, please ensure that you have specified whether consent was suitably informed.

Since your study included minors under age 18, please ensure you have stated whether you obtained consent from parents or guardians in these cases.

Additional Editor Comments:

As identified by the comments of the reviewers, there is too much ambiguity in the manuscript as it is written. There is a definite need to clarify the data and be more precise about the data and the related discussion.

Reviewers' comments:

Reviewer's Responses to Questions

**Comments to the Author**

1. Is the manuscript technically sound, and do the data support the conclusions?

Reviewer #1: Partly

Reviewer #2: Yes

2. Has the statistical analysis been performed appropriately and rigorously? 

Reviewer #1: Yes

Reviewer #2: Yes

3. Have the authors made all data underlying the findings in their manuscript fully available?

Reviewer #1: Yes

Reviewer #2: No

4. Is the manuscript presented in an intelligible fashion and written in standard English?

Reviewer #1: Yes

Reviewer #2: Yes

5. Review Comments to the Author

Reviewer #1: This very basic study compared BCG and beta-glucan-induced cytokine expression by monocytes cultured from cord blood to monocytes from adult peripheral blood.

The cytokine expression by monocytes in response to BCG exposure was very modest and IL-6 was increased only. The title of this article states focuses only on the BCG response; however my opinion is that more data are required to convincingly show that these cord blood monocytes have been 'trained' (eg metabolic changes/epigenetic changes) beyond a very modest increase in a single cytokine.

Regarding the authors' conclusions: Are cord blood monocytes phenotypically equivalent to neonatal peripheral blood monocytes? The authors imply this is the case.

Why were other cytokines not upregulated by BCG exposure (as in previous publications from the Netea group?).

Had any of the adult volunteers been BCG vaccinated?

Reviewer #2: Authors describe an experiment evaluating the cytokine produced after LPS stimulation adherent cells from cord blood and adult venous blood that were either non-primed, or primed with BCG or B-glucan. Both BCG and B-glucan can "train" innate immune cells, essentially the response to a secondary stimulus (typically a TLR agonist) is increased in trained cells over non-trained cells. Much of the data to date has focused on adults, and maybe children, the ability of mononuclear cells (primarily monocytes) to undergo innate training has not been demonstrated. Use of cord blood provides one mechanism to evaluate neonate monocyte ability to undergo training. Overall, the intent of the study was a single question, and with some improvements to writing, describes training potential of neonatal cells.

Comments:

More differences were observed with B-glucan than BCG, yet the title only mentions BCG, and discussion is focused on BCG. Inclusion of B-glucan into the title, abstract, intro, and discussion are warranted. The b-glucan findings are just as important.

Some revision for verb tense is warranted. New data should be past tense, and established (ie, published data) is present tense.

Line 69 references children and adults, line 74 only refers to adults. Is data from children the better comparison? Perhaps best to clearly define by age neonate to children?

Line 116 – is it a company or a lab that provided the BCG? It only reads as TN, USA which is unclear.

Line 129 – define the negative control – you mean RPMI-only wells? Same for Line 132

Graph titles may be better presented at top of graph, versus below. Can the cytokine name be moved to the top (as opposed to below?).

Line 148 – given how the data is presented (expressed as fold change) the use of the words “remained stable” are misleading. Perhaps it’s “were not different” instead, as remained stable suggests the cytokine was produced (and perhaps it was, but with normalization to RPMI-only group, that cannot be visualized.

Expand on line 150 to include name of all three cytokines.

Line 144 and 161 warrant revision for clarity – the response isn’t to BCG or B-glucan, it’s to LPS after prior BCG or B-glucan exposure. Inclue the work training or expand to say “…to LPS following BCG or B-glucan priming”

Line 170 – this means you did statistics between the adult and neonate for each respective treatment?

Line 181 – that data is not shown. The response to LPS would be in the RPMI only group or the RPMI+LPS group? As a reader, we cannot appreciate the response of CBAM to LPS because that was used as the normalizer. Perhaps it’s an important point in general if the same amount of IL6 is produced by non-trained cells after LPS stimulation. The sentence warrants revision for clarity. This is also true for the following sentence into line 182 – CBAM response to LPS can be maintained following initial training – again, left to infer that cytokine was produced, but because of the normalization of the data, that is not apparent. We just know it wasn’t different between non-trained (RPMI only) and BCG primed (or trained) cells.

Line 192 – can the authors try to explain why there may be a difference? It’s nice they can contrast to prior studies, but why is this study different? Was the source of BCG different between the two studies?

Line 210 warrants a citation or a profile of CBAM – what percentage of what remains in the dish is expected or shown to be other cell types that would not be present using adult venous blood?

Most of the data coming from the mouse world shows such tight error bars and similar responses with training. The human data set has greater differences, acknowledgement in the discussion is warranted. Is this genetics, or exposure to immune altering compounds in utero?

6. PLOS authors have the option to publish the peer review history of their article (what does this mean?). If published, this will include your full peer review and any attached files.

Reviewer #1: No

Reviewer #2: No

---

## [Author Response · Author response to Decision Letter 0]

5 Oct 2019

Responses to Reviewer’s comments regarding the submitted manuscript entitled “Monocytes from neonates and adults have a similar capacity to elicit trained immunity responses to BCG”.

Thank you very much for the positive assessment and useful suggestions to our submitted manuscript. Please find responses to the Reviewer’s comments and changes in the manuscript have been effected. 

We have revised the manuscript according to the PLOS ONE style templates as shown in the websites provided. 

2. Please provide additional details regarding participant consent. In the ethics statement in the Methods and online submission information, please ensure that you have specified whether consent was suitably informed. Since your study included minors under age 18, please ensure you have stated whether you obtained consent from parents or guardians in these cases. 

We obtained informed consent from the pregnant mothers. Eligibility requirements included pregnant mothers who were to deliver by cesarean section. The mothers were approached and the study purpose explained to them in a language they could understand. Those who gave informed consent on behalf of their newborn babies were recruited in the study. This has now been more clearly described in the revised manuscript.

We shall provide the repository information for our data at acceptance. All data from the present study will be available as an Excel file from the NSD database (http://nsddata.nsd.uib.no with a doi:dx.doi.org/10.18712/NSD-NSD2722-V1), starting from the date of publication of this article.

Additional Editor Comments:

As identified by the comments of the reviewers, there is too much ambiguity in the manuscript as it is written. There is a definite need to clarify the data and be more precise about the data and the related discussion.

Thank you for this comment. In the revised version, we have been more precise about the data and the discussion.

Reviewers' comments:

 Reviewer's Responses to Questions

 Comments to the Author

1. Is the manuscript technically sound, and do the data support the conclusions?

 Reviewer #1: Partly

 Reviewer #2: Yes

 2. Has the statistical analysis been performed appropriately and rigorously?

 Reviewer #1: Yes

 Reviewer #2: Yes

3. Have the authors made all data underlying the findings in their manuscript fully available?

 The PLOS Data policyhttp://www.plosone.org/static/policies.action#sharing requires authors to make all data underlying the findings described in their manuscript fully available without restriction, with rare exceptions (please refer to the Data Availability Statement in the manuscript PDF file). The data should be provided as part of the manuscript or its supporting information, or deposited to a public repository. For example, in addition to summary statistics, the data points behind means, medians and variance measures should be available. If there are restrictions on publicly sharing data—e.g. participant privacy or use of data from a third party—those must be specified. 

Reviewer #1: Yes

Reviewer #2: No

4. Is the manuscript presented in an intelligible fashion and written in standard English?

Reviewer #1: Yes

Reviewer #2: Yes

5. Review Comments to the Author

Reviewer #1: This very basic study compared BCG and beta-glucan-induced cytokine expression by monocytes cultured from cord blood to monocytes from adult peripheral blood.

The cytokine expression by monocytes in response to BCG exposure was very modest and IL-6 was increased only. The title of this article states focuses only on the BCG response; however my opinion is that more data are required to convincingly show that these cord blood monocytes have been 'trained' (e.g. metabolic changes/epigenetic changes) beyond a very modest increase in a single cytokine.

Regarding the authors' conclusions: Are cord blood monocytes phenotypically equivalent to neonatal peripheral blood monocytes? The authors imply this is the case. 

We agree that we cannot be completely sure that cord blood monocytes are equivalent to neonatal monocytes. We did not identify any studies that have examined this question, which should be investigated in future studies. We have included a cautionary remark in the Discussion on Page 12 and paragraph 2 in the revised manuscript with track changes to acknowledge this.

Why were other cytokines not upregulated by BCG exposure (as in previous publications from the Netea group?)

We note that TNF and IL-10 cytokines did not show significant increase following BCG exposure in neither adults nor neonates in our study. 

We, however, acknowledge that both IL-6 and TNF levels were increased in monocytes isolated from anonymous blood donors following BCG training in a similar in vitro assay in Kleinnijenhuis 2012 (1) and Bekkering 2016 (2). In the Bekkering paper, IL-6 responses were about 5 fold higher after 24h BCG training and later LPS restimulation, while TNF responses were about 2-fold higher. IL-10 responses did not reach significance. Encouraged by the above question we have from reviewers 1 and 2 we have added a paragraph in the manuscript discussing whether a less reactive BCG strain (BCG Bulgaria) was used in the present study. In both Kleinnijenhuis 2012 and Bekkering 2016, the BCG Denmark vaccine strain was used. Although it has not been tested for potential differences in inducing trained immunity, BCG Denmark has been shown to be more reactive and inducing higher cytokine levels upon homologous and heterologous stimulation (Anderson 2012, Ritz 2012) (3,4) This is likely the reason for weaker responses to BCG in both neonate and adults in our study, with only IL-6 being significant. 

The reason for the use of BCG Bulgaria strain in the current study was represented by the production problems at SSI Denmark during the performance of the study, with BCG-Denmark strain not being available between 2016-2019.

Had any of the adult volunteers been BCG vaccinated?

For 27 adult volunteers vaccination status was known. 22 had been vaccinated, all of them many years ago, (until 2005 Norwegians were vaccinated at around age 10-14 years). 5 adults were not vaccinated. Responses in vaccinated and unvaccinated were generally similar, with the small group of 5 non-vaccinated tending to have slightly higher median responses than the vaccinated. With the long time since vaccination and similar responses in vaccinated and unvaccinated volunteers, we believe vaccination status is unlikely to have influenced results. 

Reviewer #2: Authors describe an experiment evaluating the cytokine produced after LPS stimulation adherent cells from cord blood and adult venous blood that were either non-primed or primed with BCG or B-glucan. Both BCG and B-glucan can "train" innate immune cells, essentially the response to a secondary stimulus (typically a TLR agonist) is increased in trained cells over non-trained cells. Much of the data to date has focused on adults, and maybe children, the ability of mononuclear cells (primarily monocytes) to undergo innate training has not been demonstrated. The use of cord blood provides one mechanism to evaluate neonate monocyte ability to undergo training.

Overall, the intent of the study was a single question, and with some improvements to writing, describes training potential of neonatal cells.

 Comments:

More differences were observed with B-glucan than BCG, yet the title only mentions BCG, and discussion is focused on BCG. Inclusion of B-glucan into the title, abstract, intro, and discussion are warranted. The b-glucan findings are just as important. Some revision for verb tense is warranted. New data should be past tense, and established (ie, published data) is present tense.

We agree. We have now included β-glucan in the title, abstract, intro and added discussion of these findings in the discussion as advised and the revision for verb tense has been done.

Line 69 references children and adults, line 74 only refers to adults.

Line 69 refers to some of the studies that have demonstrated trained innate immunity in both children and adults. However, line 74 describes the in vitro induction of trained immunity model which was studied in adults but not neonates, hence the reason for us to study this model in neonates. 

Is data from children the better comparison? Perhaps best to clearly define by age neonate to children?

There is hardly data on trained innate immunity of adherent mononuclear cells in neonates. 

We agree that it is better to change "children" to "infants" as shown in a few studies done in infants; Smith SG et al, 2017 (5) and Jensen KJ, et al, 2015 (6). These studies however, didn't use adherent monocytes and were post-vaccination studies. 

 To be more specific we have changed the word “children” to “infants” in the introduction. 

Line 116 – is it a company or a lab that provided the BCG? It only reads as TN, USA which is unclear.

The reviewer here obviously meant β-glucan and not BCG. β-glucan was kindly provided by Professor David Williams (East Tennessee State University, USA). We have corrected this in the Methods section.

Line 129 – define the negative control – you mean RPMI-only wells?

Same for Line 132

Yes, the negative control is the RPMI- only wells. We have now defined it more clearly in the mentioned paragraphs. 

Graph titles may be better presented at top of graph, versus below. Can the cytokine name be moved to the top (as opposed to below?).

We thank the reviewer for good suggestions. The figures have been improved accordingly.

Line 148 – given how the data is presented (expressed as fold change) the use of the words “remained stable” are misleading. Perhaps it’s “were not different” instead, as remained stable suggests the cytokine was produced (and perhaps it was, but with normalization to RPMI-only group, that cannot be visualized.

Yes, we agree that “were not different” is a clearer formulation. This has been corrected in the revised manuscript. 

Expand on line 150 to include name of all three cytokines.

Naming of all three cytokines has been included. 

Line 144 and 161 warrant revision for clarity – the response isn’t to BCG or B-glucan, it’s to LPS after prior BCG or B-glucan exposure. Include the word training or expand to say “…to LPS following BCG or B-glucan priming”

We agree. This has been corrected to include “to LPS following BCG or β-glucan priming”

Line 170 – this means you did statistics between the adult and neonate for each respective treatment?

Yes, to help make this clearer we rephrased sentences about comparison between adults and neonates. 

Line 181 – that data is not shown. The response to LPS would be in the RPMI only group or the RPMI+LPS group? As a reader, we cannot appreciate the response of CBAM to LPS because that was used as the normalizer. Perhaps it’s an important point in general if the same amount of IL6 is produced by non-trained cells after LPS stimulation. The sentence warrants revision for clarity. 

We agree that this sentence can be better formulated to reflect the presented data. The production of cytokines in the BCG and β-glucan primed cells that were restimulated with LPS were normalized against the cytokine production in the non-primed (RPMI only) LPS-restimulated cells. For more information: there was cytokine production in the non-primed cell cultures used for normalization, and this was similar in adults and neonates; i.e. for any of the three cytokines, measured levels of non-primed, LPS-restimulated monocytes were not significantly different in adult and neonates. 

In the revised manuscript, the inserted sentence “and that CBAM respond to LPS re-stimulation” has been removed for clarity.

This is also true for the following sentence into line 182 – CBAM response to LPS can be maintained following initial training – again, left to infer that cytokine was produced, but because of the normalization of the data, that is not apparent. We just know it wasn’t different between non-trained (RPMI only) and BCG primed (or trained) cells.

We agree and have clarified the sentence to “Our data shows that some CBAM cytokine responses, when restimulated with LPS following initial training with either BCG or β-glucan stimuli, are increased compared to LPS-restimulated non-trained CBAM.” as this is what we want to show by calculating and presenting our data as fold change. 

Line 192 – can the authors try to explain why there may be a difference? It’s nice they can contrast to prior studies, but why is this study different? Was the source of BCG different between the two studies?

We note that TNF and IL-10 cytokines did not show significant increase following BCG exposure in neither adults nor neonates in our study. 

We, however, acknowledge that both IL-6 and TNF levels were increased in monocytes isolated from anonymous blood donors following BCG training in a similar in vitro assay in Kleinnijenhuis 2012 (1) and Bekkering 2016 (2). In the Bekkering paper, IL-6 responses were about 5 fold higher after 24h BCG training and later LPS restimulation, while TNF responses were about 2-fold higher. IL-10 responses did not reach significance. Encouraged by the above question we have from reviewers 1 and 2 we have added a paragraph in the manuscript discussing whether a less reactive BCG strain (BCG Bulgaria) was used in the present study. In both Kleinnijenhuis 2012 and Bekkering 2016, the BCG Denmark vaccine strain was used. Although it has not been tested for potential differences in inducing trained immunity, BCG Denmark has been shown to be more reactive and inducing higher cytokine levels upon homologous and heterologous stimulation (Anderson 2012, Ritz 2012) (3,4) This is likely the reason for weaker responses to BCG in both neonate and adults in our study, with only IL-6 being significant. The reason for the use of BCG Bulgaria strain in the current study was represented by the production problems at SSI Denmark during the performance of the study, with BCG-Denmark strain not being available between 2016-2019.

Another difference was use of different concentrations of BCG. In Kleinnijenhuis 2012 and Bekkering 2016 studies, they used 1µg/mL and 10µg/mL, respectively, while in the current study we used 5µg/mL.

Line 210 warrants a citation or a profile of CBAM – what percentage of what remains in the dish is expected or shown to be other cell types that would not be present using adult venous blood? Most of the data coming from the mouse world shows such tight error bars and similar responses with training. The human data set has greater differences, acknowledgement in the discussion is warranted. Is this genetics, or exposure to immune altering compounds in utero?

We agree that this is a fair discussion point. A profile of CBAM in a study by Normann Erik et al, showed a purification of non‐active, but stimulation‐competent monocytes with high yields (2.3–9×107 cells) and purity of 70–90% (average 83%) by adherence or 90–98% by flow cytometric analysis (7).

We have added this issue and the reference in the Discussion.

6. PLOS authors have the option to publish the peer review history of their article (what does this mean?https://journals.plos.org/plosone/s/editorial-and-peer-review-process#loc-peer-review-history). If published, this will include your full peer review and any attached files.

Do you want your identity to be public for this peer review? For information about this choice, including consent withdrawal, please see our Privacy Policyhttps://www.plos.org/privacy-policy.

Reviewer #1: No

Reviewer #2: No

Please log into your account, locate the manuscript record, and check for the action link "View Attachments". If this link does not appear, there are no attachment files to be viewed.]

While revising your submission, please upload your figure files to the Preflight Analysis and Conversion Engine (PACE) digital diagnostic tool, https://pacev2.apexcovantage.com/. PACE helps ensure that figures meet PLOS requirements. To use PACE, you must first register as a user. Registration is free. Then, login and navigate to the UPLOAD tab, where you will find detailed instructions on how to use the tool. If you encounter any issues or have any questions when using

PACE, please email us at figures@plos.org<mailto:figures@plos.org>.

Please note that Supporting Information files do not need this step.

References

1. Kleinnijenhuis J, Quintin J, Preijers F, et al. Bacille Calmette-Guerin induces NOD2-dependent nonspecific protection from reinfection via epigenetic reprogramming of monocytes. Proc Natl Acad Sci U S A 2012; 109:17537-42.

2. Bekkering S, Blok BA, Joosten LA, Riksen NP, van Crevel R, Netea MG. In Vitro Experimental Model of Trained Innate Immunity in Human Primary Monocytes. Clin Vaccine Immunol 2016; 23:926-33.

3. Anderson EJ WE, Mawa PA, Kizza M, Lyadda N, Nampijja M, Elliott AM. The influence of BCG vaccine strain on mycobacteria-specific and non-specific immune responses in a prospective cohort of infants in Uganda. Vaccine. 2012;30(12):2083-9.

4. Ritz N, Dutta B, Donath S, Casalaz D, Connell TG, Tebruegge M, et al. The influence of bacille Calmette-Guerin vaccine strain on the immune response against tuberculosis: a randomized trial. Am J Respir Crit Care Med. 2012;185(2):213-22.

5. Smith SG, Kleinnijenhuis J, Netea MG, Dockrell HM. Whole Blood Profiling of Bacillus Calmette-Guerin-Induced Trained Innate Immunity in Infants Identifies Epidermal Growth Factor, IL-6, Platelet-Derived Growth Factor-AB/BB, and Natural Killer Cell Activation. Frontiers in immunology. 2017;8:644.

6. Jensen KJ, Larsen N, Biering-Sorensen S, Andersen A, Eriksen HB, Monteiro I, et al. Heterologous immunological effects of early BCG vaccination in low-birth-weight infants in Guinea-Bissau: a randomized-controlled trial. J Infect Dis. 2015;211(6):956-67.

7. Normann E, Lacaze-Masmonteil T, Winkler-Lowen B, Guilbert LJ. Isolation of non-activated monocytes from human umbilical cord blood. Am J Reprod Immunol 2010; 63:66-72.

---

## [Decision Letter · Decision Letter 1]

5 Nov 2019

PONE-D-19-19049R1

Monocytes from neonates and adults have a similar capacity to elicit trained immunity responses to BCG and β-glucan

PLOS ONE

Dear Dr Hanevik,

Thank you for resubmitting your manuscript to PLOS ONE. After careful consideration, we feel that it has merit but does not fully meet PLOS ONE’s publication criteria as it currently stands. Therefore, we invite you to submit a revised version of the manuscript that addresses the points raised during the review process. Based on the reviewer comments, some issues were not directly addressed, thus please pay careful attention to these comments.

We would appreciate receiving your revised manuscript by Dec 20 2019 11:59PM. To enhance the reproducibility of your results, we recommend that if applicable you deposit your laboratory protocols in protocols.io, where a protocol can be assigned its own identifier (DOI) such that it can be cited independently in the future. For instructions see: http://journals.plos.org/plosone/s/submission-guidelines#loc-laboratory-protocols

We look forward to receiving your revised manuscript.

Kind regards,

Angelo A. Izzo

Academic Editor

PLOS ONE

Reviewers' comments:

Reviewer's Responses to Questions

**Comments to the Author**

1. If the authors have adequately addressed your comments raised in a previous round of review and you feel that this manuscript is now acceptable for publication, you may indicate that here to bypass the “Comments to the Author” section, enter your conflict of interest statement in the “Confidential to Editor” section, and submit your "Accept" recommendation.

Reviewer #1: (No Response)

Reviewer #2: All comments have been addressed

2. Is the manuscript technically sound, and do the data support the conclusions?

Reviewer #1: No

Reviewer #2: (No Response)

3. Has the statistical analysis been performed appropriately and rigorously? 

Reviewer #1: Yes

Reviewer #2: (No Response)

4. Have the authors made all data underlying the findings in their manuscript fully available?

Reviewer #1: Yes

Reviewer #2: (No Response)

5. Is the manuscript presented in an intelligible fashion and written in standard English?

Reviewer #1: Yes

Reviewer #2: (No Response)

6. Review Comments to the Author

Reviewer #1: My original major concern remains: I do not feel it is sufficient to term cytokine production by monocytes 'training' without showing additional features that are characteristic of innate immune training (eg switch in metabolism to glycolysis, epigenetic changes, other functional changes such as reactive oxygen species production or the longevity of the response). The authors have not addressed this concern and should do so by re-considering their aims and conclusions. The cytokine production may indicate in vitro training, but the authors do not provide any evidence that this was the case and need to acknowledge this.

The authors should include the vaccination status of the adult volunteers in the Methods.

Title: revisit - is this really training?

Abstract: L38 - this study does not address the ability of BCG and beta-glucan to induce trained immunity in neonates. It studies the ability of BCG and beta-glucan to induce cytokine production in cord blood monocytes.

Typographical errors:

L55 - use the abbreviation TB

L154 females  female

L163 cytokine levels  cytokines (IL-6, ...)

L213 - replace the 'the'

L247 - ... therefore this should be ...

L248 The Results  The results

Tenses are still a bit confused in the discussion - established knowledge/data in the present tense, but when you refer to a specific study (published or not) use the past tense.

Reviewer #2: (No Response)

7. PLOS authors have the option to publish the peer review history of their article (what does this mean?). If published, this will include your full peer review and any attached files.

Reviewer #1: No

Reviewer #2: No

---

## [Author Response · Author response to Decision Letter 1]

14 Nov 2019

Professor Angelo A. Izzo November 11th, 2019

Editor 

PLoS ONE,

Dear Professor Izzo 

PONE-D-19-19049R1

Responses to Reviewer’s comments regarding the submitted manuscript entitled “Monocytes from neonates and adults have a similar capacity to elicit trained immunity responses to BCG”.

Thank you very much for the positive assessment to our revised submitted manuscript. Please find responses to the reviewer’s comments to our revised manuscript. The manuscript has been revised according to many of the comments. We thank the reviewers for taking the time to point out some grammatical improvements.

Comments to the Author

1. If the authors have adequately addressed your comments raised in a previous round of review and you feel that this manuscript is now acceptable for publication, you may indicate that here to bypass the “Comments to the Author” section, enter your conflict of interest statement in the “Confidential to Editor” section, and submit your "Accept" recommendation.

Reviewer #1: (No Response)

Reviewer #2: All comments have been addressed

2. Is the manuscript technically sound, and do the data support the conclusions?

Reviewer #1: No

Reviewer #2: (No Response)

3. Has the statistical analysis been performed appropriately and rigorously?

Reviewer #1: Yes

Reviewer #2: (No Response)

4. Have the authors made all data underlying the findings in their manuscript fully available?

The PLOS Data policyhttp://www.plosone.org/static/policies.action#sharing requires authors to make all data underlying the findings described in their manuscript fully available without restriction, with rare exception (please refer to the Data Availability Statement in the manuscript PDF file). The data should be provided as part of the manuscript or its supporting information, or deposited to a public repository. For example, in addition to summary statistics, the data points behind means, medians and variance measures should be available. If there are restrictions on publicly sharing data—e.g. participant privacy or use of data from a third party—those must be specified.

Reviewer #1: Yes

Reviewer #2: (No Response)

5. Is the manuscript presented in an intelligible fashion and written in standard English?

Reviewer #1: Yes

Reviewer #2: (No Response)

6. Review Comments to the Author

Reviewer #1: My original major concern remains: I do not feel it is sufficient to term cytokine production by monocytes 'training' without showing additional features that are characteristic of innate immune training (eg switch in metabolism to glycolysis, epigenetic changes, other functional changes such as reactive oxygen species production or the longevity of the response). The authors have not addressed this concern and should do so by re-considering their aims and conclusions. The cytokine production may indicate in vitro training, but the authors do not provide any evidence that this was the case and need to acknowledge this.

Answer. This is an important point that is raised by the reviewer. Strictly, the term of trained immunity depicts the long-term capacity of innate immune cells to over-respond to a second stimulation, after the short-term exposure to an initial stimulus. The marker of that is generally used is cytokine production, while the mechanisms behind it are metabolic and epigenetic. This has been shown by measuring cytokines in a number of studies, and the in vitro model we use it has been described for adults. 

We have now used the same model in cord blood monocytes. The experimental method used in this study is an in vitro model of trained immunity published in several papers (Kleinnijenhuis et al., 2012, Arts et al 2016 and Bekkering et al 2016). In addition to measuring cytokine responses as proof of training, the mentioned studies also used the model to investigate the metabolic and epigenetic mechanisms related to trained immunity. Although this would have been interesting as added measurements of trained immunity, our focus in the present study was to compare adult and cord blood monocyte trained immune responses in a well-established model with the commonly used immunological readouts of trained immunity. Not enough material is unfortunately currently available to extend these studies to metabolic and epigenetic parameters.

We have addressed the reviewers concern by stating in the discussion that epigenetic and metabolic changes to further define a ‘broad trained immunity phenotype’ was not done in this study. In the immunological sense, showing hyper-responsiveness is enough to demonstrate trained immunity.

The authors should include the vaccination status of the adult volunteers in the Methods.

Answer. We have now included vaccination status of adults in methods section of the revised manuscript.

Title: revisit - is this really training?

Answer. Based on the arguments exposed above, we argue that the process studied in this ex-vivo model is indeed training, both by definition and by readouts in an established model for trained immunity. 

Abstract: L38 - this study does not address the ability of BCG and beta-glucan to induce trained immunity in neonates. It studies the ability of BCG and beta-glucan to induce cytokine production in cord blood monocytes.

Answer. In our opinion “the ability of BCG and beta-glucan to induce cytokine production in cord blood monocytes”, when restimulated with LPS after a resting period is per definition a trained immunity response as seen in this model in several studies and the way this term is used in the current literature. We do not merely study the direct responsiveness of cord blood monocytes to acute stimulation. We therefore believe the text in the abstract reflects this correctly. 

However, we agree that replacing the term “neonates” with “cord blood monocytes” in this sentence states the aim in a more specific way, and this has been replaced in the abstract.

Typographical errors:

L55 - use the abbreviation TB 

The abbreviation ‘TB’ has been used instead of ‘tuberculosis’ in the revised manuscript.

L154 females  female 

This error has been corrected.

L163 cytokine levels  cytokines (IL-6, ...) 

The cytokines (IL-6, IL-10 and TNF) have been mentioned in the revised manuscript. 

L213 - replace the 'the' 

‘the’ has been replaced in the manuscript

L247 - ... therefore this should be ... 

The sentence has been revised in the manuscript

L248 The Results  The results 

This has been corrected in the manuscript

Tenses are still a bit confused in the discussion - established knowledge/data in the present tense, but when you refer to a specific study (published or not) use the past tense.

Thanks for this observation; tenses in the discussion have been revised once again in the revised manuscript

Reviewer #2: (No Response)

7. PLOS authors have the option to publish the peer review history of their article (what does this mean?https://journals.plos.org/plosone/s/editorial-and-peer-review-process#loc-peer-review-history). If published, this will include your full peer review and any attached files

Do you want your identity to be public for this peer review? For information about this choice, including consent withdrawal, please see our Privacy Policyhttps://www.plos.org/privacy-policy.

Reviewer #1: No

Reviewer #2: No

While revising your submission, please upload your figure files to the Preflight Analysis and Conversion Engine (PACE) digital diagnostic tool, https://pacev2.apexcovantage.com/. PACE helps ensure that figures meet PLOS requirements. To use PACE, you must first register as a user. Registration is free. Then, login and navigate to the UPLOAD tab, where you will find detailed instructions on how to use the tool. If you encounter any issues or have any questions when using PACE, please email us at figures@plos.org<mailto:figures@plos.org>. Please note that Supporting Information files do

---

## [Decision Letter · Decision Letter 2]

18 Dec 2019

PONE-D-19-19049R2

Monocytes from neonates and adults have a similar capacity to elicit trained immunity responses to BCG and β-glucan

PLOS ONE

Dear Dr Hanevik,

Thank you for submitting your manuscript to PLOS ONE. After careful consideration, we feel that it has merit but does not fully meet PLOS ONE’s publication criteria as it currently stands. Therefore, we invite you to submit a revised version of the manuscript that addresses the points raised during the review process.

We would appreciate receiving your revised manuscript by Feb 01 2020 11:59PM. To enhance the reproducibility of your results, we recommend that if applicable you deposit your laboratory protocols in protocols.io, where a protocol can be assigned its own identifier (DOI) such that it can be cited independently in the future. For instructions see: http://journals.plos.org/plosone/s/submission-guidelines#loc-laboratory-protocols

We look forward to receiving your revised manuscript.

Kind regards,

Angelo A. Izzo

Academic Editor

PLOS ONE

Reviewers' comments:

Reviewer's Responses to Questions

**Comments to the Author**

1. If the authors have adequately addressed your comments raised in a previous round of review and you feel that this manuscript is now acceptable for publication, you may indicate that here to bypass the “Comments to the Author” section, enter your conflict of interest statement in the “Confidential to Editor” section, and submit your "Accept" recommendation.

Reviewer #1: (No Response)

2. Is the manuscript technically sound, and do the data support the conclusions?

Reviewer #1: Partly

3. Has the statistical analysis been performed appropriately and rigorously? 

Reviewer #1: Yes

4. Have the authors made all data underlying the findings in their manuscript fully available?

Reviewer #1: Yes

5. Is the manuscript presented in an intelligible fashion and written in standard English?

Reviewer #1: Yes

6. Review Comments to the Author

Reviewer #1: The authors strongly argue that what they observe (increased cytokine production) is trained immunity. That may be true; however I still have concerns about this, not least because the responses are quite different (perhaps due to the BCG strain used) from previously published data. This could mean that the 'training' (ie epigenetic/metabolic changes etc) observed in the earlier published work did not occur here. In order to feel satisfied accepting this I think the authors should restrain their conclusions (ie in the abstract instead of saying "BCG and B-glucan induce a similar level of trained immunity in neonatal monocytes" they should more carefully state "BCG and B-glucan induce increased cytokine production, reminiscent of trained immunity, in neonatal monocytes". I understand it is not possible to do further testing on the samples; however simply measuring cytokines is insufficient to indicate trained immunity, especially when the results are different (diminished) compared to the published work using this model.

7. PLOS authors have the option to publish the peer review history of their article (what does this mean?). If published, this will include your full peer review and any attached files.

Reviewer #1: No

---

## [Author Response · Author response to Decision Letter 2]

23 Dec 2019

PONE-D-19-19049R2

Responses to comment from Reviewer#1 regarding the resubmitted manuscript entitled “Monocytes from neonates and adults have a similar capacity to elicit trained immunity responses to BCG”.

Reviewer #1: The authors strongly argue that what they observe (increased cytokine production) is trained immunity. That may be true; however I still have concerns about this, not least because the responses are quite different (perhaps due to the BCG strain used) from previously published data. This could mean that the 'training' (ie epigenetic/metabolic changes etc) observed in the earlier published work did not occur here.

In order to feel satisfied accepting this I think the authors should restrain their conclusions (ie in the abstract instead of saying "BCG and B-glucan induce a similar level of trained immunity in neonatal monocytes" they should more carefully state "BCG and B-glucan induce increased cytokine production, reminiscent of trained immunity, in neonatal monocytes".

I understand it is not possible to do further testing on the samples; however simply measuring cytokines is insufficient to indicate trained immunity, especially when the results are different (diminished) compared to the published work using this model.

Reply: We agree we could be a little more cautious in the interpretation and have changed the relevant sentences in conclusions in the abstract and manuscript in line with the reviewer’s request. 

To further underline the issue raised by the reviewer we also added the sentence “Including measurement of epigenetic and metabolic alterations in such experiments will be helpful to characterize the nature of the effects and mechanisms behind them“ in the discussion about further experiments.

---

## [Decision Letter · Decision Letter 3]

14 Jan 2020

PONE-D-19-19049R3

Monocytes from neonates and adults have a similar capacity to elicit trained immunity responses to BCG and β-glucan

PLOS ONE

Dear Dr Hanevik,

Thank you for submitting your manuscript to PLOS ONE. After careful consideration, we feel that it has merit but does not fully meet PLOS ONE’s publication criteria as it currently stands. Therefore, we invite you to submit a revised version of the manuscript that addresses the points raised during the review process.

We would appreciate receiving your revised manuscript by Feb 28 2020 11:59PM. To enhance the reproducibility of your results, we recommend that if applicable you deposit your laboratory protocols in protocols.io, where a protocol can be assigned its own identifier (DOI) such that it can be cited independently in the future. For instructions see: http://journals.plos.org/plosone/s/submission-guidelines#loc-laboratory-protocols

We look forward to receiving your revised manuscript.

Kind regards,

Angelo A. Izzo

Academic Editor

PLOS ONE

Additional Editor Comments (if provided):

I strongly recommend that you read the comments made by Reviewer 1 and make the changes requested.

Reviewers' comments:

Reviewer's Responses to Questions

**Comments to the Author**

1. If the authors have adequately addressed your comments raised in a previous round of review and you feel that this manuscript is now acceptable for publication, you may indicate that here to bypass the “Comments to the Author” section, enter your conflict of interest statement in the “Confidential to Editor” section, and submit your "Accept" recommendation.

Reviewer #1: (No Response)

2. Is the manuscript technically sound, and do the data support the conclusions?

Reviewer #1: Partly

3. Has the statistical analysis been performed appropriately and rigorously? 

Reviewer #1: Yes

4. Have the authors made all data underlying the findings in their manuscript fully available?

Reviewer #1: Yes

5. Is the manuscript presented in an intelligible fashion and written in standard English?

Reviewer #1: Yes

6. Review Comments to the Author

Reviewer #1: The authors have made some quite minor amendments to the manuscript to address my concern, which is that I think it is incorrect to use the term 'trained immunity', without evidence of this beyond cytokine production particularly given that the cytokine responses observed in this study differed to those from the cited earlier publications that use the 'well-established' model.

This authors can address this by properly restraining conclusions in the title, abstract and full manuscript; however the authors have only made a small attempt to do this and in my opinion most people giving this manuscript a cursory read, as it stands, would not notice that other aspects of training (beyond cytokine production) have not been determined and it is relying on previous published work.

7. PLOS authors have the option to publish the peer review history of their article (what does this mean?). If published, this will include your full peer review and any attached files.

Reviewer #1: No

---

## [Author Response · Author response to Decision Letter 3]

22 Jan 2020

Reviewer #1: The authors have made some quite minor amendments to the manuscript to address my concern, which is that I think it is incorrect to use the term 'trained immunity', without evidence of this beyond cytokine production particularly given that the cytokine responses observed in this study differed to those from the cited earlier publications that use the 'well-established' model.

This authors can address this by properly restraining conclusions in the title, abstract and full manuscript; however the authors have only made a small attempt to do this and in my opinion most people giving this manuscript a cursory read, as it stands, would not notice that other aspects of training (beyond cytokine production) have not been determined and it is relying on previous published work

Response: In the previous revision, we introduced the changed requested by the reviewer. When it is now requested to make more thorough revision of the use of the term “trained immunity” we have discussed this between us and acknowledge that there are different opinions regarding the use of this term. 

In addition to the already introduced cautious wording “reminiscent of trained immunity”, as advised by the reviewer previously, we have therefore now revised the manuscript to focus on presenting the cytokine response data. We have used more cautious wording and made amendments to the revised manuscript with properly restraining conclusions regarding “trained immunity” in the title, abstract and full manuscript.

---

## [Decision Letter · Decision Letter 4]

4 Feb 2020

Monocytes from neonates and adults have a similar capacity to adapt their cytokine production after previous exposure to BCG and β-glucan

PONE-D-19-19049R4

Dear Dr. Hanevik,

We are pleased to inform you that your manuscript has been judged scientifically suitable for publication and will be formally accepted for publication once it complies with all outstanding technical requirements.

With kind regards,

Angelo A. Izzo

Academic Editor

PLOS ONE

Additional Editor Comments (optional):

Reviewers' comments:

Reviewer's Responses to Questions

**Comments to the Author**

1. If the authors have adequately addressed your comments raised in a previous round of review and you feel that this manuscript is now acceptable for publication, you may indicate that here to bypass the “Comments to the Author” section, enter your conflict of interest statement in the “Confidential to Editor” section, and submit your "Accept" recommendation.

Reviewer #1: All comments have been addressed

2. Is the manuscript technically sound, and do the data support the conclusions?

Reviewer #1: (No Response)

3. Has the statistical analysis been performed appropriately and rigorously? 

Reviewer #1: (No Response)

4. Have the authors made all data underlying the findings in their manuscript fully available?

Reviewer #1: (No Response)

5. Is the manuscript presented in an intelligible fashion and written in standard English?

Reviewer #1: (No Response)

6. Review Comments to the Author

Reviewer #1: (No Response)

7. PLOS authors have the option to publish the peer review history of their article (what does this mean?). If published, this will include your full peer review and any attached files.

Reviewer #1: No

---

## [Editor Report · Acceptance letter]

10 Feb 2020

PONE-D-19-19049R4 

Monocytes from neonates and adults have a similar capacity to adapt their cytokine production after previous exposure to BCG and β-glucan 

Dear Dr. Hanevik:

I am pleased to inform you that your manuscript has been deemed suitable for publication in PLOS ONE. Congratulations! Your manuscript is now with our production department. 

With kind regards,

on behalf of

Dr. Angelo A. Izzo 

Academic Editor

PLOS ONE